# Effects of GrandFusion Diet on Cognitive Impairment in Transgenic Mouse Model of Alzheimer’s Disease

**DOI:** 10.3390/nu13010117

**Published:** 2020-12-30

**Authors:** Jin Yu, Hong Zhu, Saeid Taheri, William Mondy, Stephen Perry, Cheryl Kirstein, Mark S. Kindy

**Affiliations:** 1Department of Pharmaceutical Sciences, College of Pharmacy, University of South Florida, Tampa, FL 33620, USA; jinyu@usf.edu (J.Y.); hongzhu@usf.edu (H.Z.); taheris@usf.edu (S.T.); wmondy@usf.edu (W.M.); 2NutriFusion®, LLC, Naples, FL 34109, USA; sperry@consealint.com; 3Department of Psychology, College of Arts and Sciences, University of South Florida, Tampa, FL 33620, USA; kirstein@usf.edu; 4James A. Haley Veterans Administration Medical Center, Research, Tampa, FL 33612, USA; 5Shriners Hospital for Children, Research, Tampa, FL 33612, USA

**Keywords:** diet, Alzheimer’s disease, inflammation, behavior, amyloid

## Abstract

Alzheimer’s disease (AD) is the result of the deposition of amyloid β (Aβ) peptide into amyloid fibrils and tau into neurofibrillary tangles. At the present time, there are no possible treatments for the disease. We have recently shown that diets enriched in phytonutrients show protection or limit the extent of damage in a number of neurological disorders. GrandFusion (GF) diets have attenuated the outcomes in animal models of traumatic brain injury, cerebral ischemia, and chronic traumatic encephalopathy. In this study, we investigated the effect of GF diets in a mouse model of AD prior to the development of amyloid plaques to show how this treatment paradigm would alter the accumulation of Aβ peptide and related pathologic changes (i.e., inflammation, cathepsin B, and memory impairment). Administration of GF diets (2–4%) over a period of four months in APP/ΔPS1 double-transgenic mice resulted in attenuation in Aβ peptide levels, reduction of amyloid load, and inflammation, increased cathepsin B expression, and improved spatial orientation. Additionally, treatment with GF diets increased nerve growth factor (NGF) levels in the brain and tempered the memory impairment in the animal model. These data suggest that GF diets may alter the development and progression of the mechanisms associated with the disease process to effectively modify AD pathogenesis.

## 1. Introduction

Alzheimer’s disease (AD) impacts more than 5.7 million people in the United States alone and is one of the most common forms of dementia and the rate and numbers are expected to increase due to the aging population [1]. Approximately 1 in 10 individuals over the age of 65 have AD, and that ratio increases to 1 in 3 after the age of 85. Unfortunately, diagnosis of the disease does not occur until 5–8 years after disease onset, and death usually occurs within 3–9 years after the onset of symptoms [2]. AD is elaborated by both clinical and pathological endpoints that manifest in cognitive deficits and other neurological changes that appear to be irreversible [2,3]. Furthermore, AD is characterized by the presence of aggregated amyloid-β (Aβ) peptides that result in extracellular amyloid plaques and hyperphosphorylated tau protein that accumulates within the neurons as neurofibrillary tangles (NFTs). The Aβ peptides result from the progressive cleavage of the amyloid precursor protein (APP) by β- and γ-secretases giving rise to increased Aβ_1–40_ and Aβ_1–42_ peptides, which exacerbates the aggregation and, eventually, amyloid plaques [4]. Previous studies have shown that aggregates of Aβ peptides are neurotoxic and perform an important role in the pathogenesis of AD and accumulate due to both decreased turnover and increased production [5,6].

Numerous studies over the years have implicated diets or diet supplements as having an impact on the outcomes associated with AD [7,8]. Most have been performed in the various APP animal models, and the correlation to AD has been tenuous but substantial [9]. Meta-analyses of human studies have suggested that antioxidant-rich foods (diets with nuts, grapes, cherries) and certain fatty acid supplementation, particularly with n-3 polyunsaturated fatty acids (PUFAs), that specific cognitive functions and cognitive-related outcomes in mild cognitive impairment (MCI), mild-to-moderate dementia, and AD, were improved [10,11]. In addition, antioxidant vitamins and supplementation with trace elements improved specific cognitive-related outcomes and biomarkers, and high-dose vitamin B in AD and MCI patients improved cognitive outcomes, while folic acid had a positive impact on specific cognitive domains in those with elevated homocysteine [12,13]. Therefore, supplementation of diets with specific nutrients may help to attenuate not only aging but AD.

GrandFusion^®^ (GF) diets are a blend of various fruits and vegetables, that are prepared with enhanced vitamins and nutrients that are able to limit the extent of cerebral ischemia injury and attenuate several parameters of stroke, specifically inflammatory markers, reactive oxygen species (ROS) and behavioral changes [14]. Moreover, GF diets have been shown to improve memory and learning in aged rats and increase physical activity as assessed by antioxidant enzymes and signaling pathways [15]. Our studies have demonstrated that GF diets have a multitude of activities, including anti-inflammatory, antioxidant, neuroprotective, and neurogenic properties [16,17,18].

Based on the impact of GF diets on altered inflammation and oxidative stress, we decided to test the efficacy of GF diets in a mouse model of AD. Using the APP/PS1 mouse model that we previously have documented, mice were gavaged daily with the 2 and 4% GF diets enriched with fruits and vegetables and examined the animals for the impact on neurological outcomes (behavior), biochemical changes and amyloid pathology [19]. These studies demonstrate that the GF diets, when started at the time or before amyloid deposition and continued for four months, in the transgenic mice expressing familial AD (FAD) mutant AβPP and presenilin 1 (PS1) cDNAs (AβPP-PS1), we were able to significantly attenuate amyloid plaque burden, levels of Aβ peptide, cathepsin B levels (inflammation) and increased brain NGF when compared to AβPP-PS1 control mice. These data suggest that the GF diets may perhaps be beneficial in the slowing of the AD process in man.

## 2. Materials and Methods

### 2.1. Animals

Approximately 120 4-month-old transgenic animals were used in the studies. Both male and female mice were used (10 of each for each group). The transgenic mice expressed the mutant human form of presenilin-1 (DeltaE9) and the altered form of the mouse/human amyloid precursor protein (APP695) [20,21]. The mouse prion protein (Prp) promoter directed the expression of both transgenes. The DeltaE9 mutation in the presenilin-1 gene expresses a protein with a deletion of exon 9 and relates to a form linked with early-onset AD. The APP695 gene contains the K595N/M596L (Swedish) AD-causing mutations. The coding sequence of mouse Aβ peptide domain was humanized via alterations in the three amino acids that are different between the species with the human residues. These APP/ΔPS1-Tg mice start forming amyloid-like plaques at around 3 to 4 months of age. These mice were on a C3H/HeJ*C57BL/6J background. Ten 8-month-old wild-type (C3H/HeJ*C57BL/6) mice were obtained to establish the baseline levels of the different variables.

### 2.2. Diet Treatment

Animals were assigned randomly to a vehicle group (*n* = 20) or treatment groups (*n* = 40). Four months prior to euthanasia, animals were fed a normal diet or a normal diet supplemented with 2 or 4% of the NF-216 (GrandFusion–Fruit and Veggie #1 Blend). See Appendix A for composition of supplementation. Animals were gavaged with the supplements on a daily basis, once per day. Supplements (GrandFusion) were prepared by NutriFusion, LLC (www.nutrifusion.com). Average food intake was 3.82 ± 0.09 g/day/mouse, and the average diet consumption was 0.08 ±  0.006 g/day/mouse. There were no differences in food intake between the groups. All studies were submitted and approved by the Institutional Animal Care and Use Committee (IACUC) at the Medical University of South Carolina (MUSC) and the Ralph H. Johnson Veterans Affairs Medical Center (VAMC). The approved study adhered to the Guide for the Care and Use of Laboratory Animals developed by the Office of Laboratory Animal Welfare (OLAW). At the end of the studies, brains (one hemisphere) were immersion-fixed in 4% paraformaldehyde for 24 h, followed by immersion in 4% paraformaldehyde containing 30% sucrose for 2 to 3 days. Following fixation, the brains were frozen in OCT medium and sectioned on a cryostat to obtain 30-μm frozen sections for immunohistochemical analysis. The other brain hemispheres were rapidly frozen and used to quantitate the levels of Aβ peptide (Aβ_1–40_ and Aβ_1–42_), inflammatory markers, NGF, and cathepsin B. Right and left brain hemispheres were randomly selected for the different studies.

### 2.3. Immunohistochemistry Staining

Cryosections of the hemispheres were rinsed three times (5 min/wash) with Tris-buffered saline (TBS) (pH 7.4) buffer, a subsequently washed one time for 5 min with 0.1% Triton X-100-TBS buffer. Sections were incubated in 3% H_2_O_2_ and TBS buffer for at least 30 min at room temperature to remove endogenous peroxidase activity. Following 1 h of blocking with around 5.0% serum (horse or goat), the sections were incubated overnight with primary antibodies. Primary antibodies dilutions were: Aβ peptide, mouse anti-human Aβ peptide antibody (1:500 dilution, 10D5; Elan Pharmaceuticals, San Francisco, CA, USA); CD68-positive microglia, anti-mouse CD68 antibody (1:400 dilution, KP1; Abcam, Cambridge, MA, USA); glial fibrillary acidic protein (GFAP)-positive astrocytes (1:200 dilution, 2E1; BD Biosciences, San Jose, CA, USA). Afterwards, sections were washed three times (5 min/wash) with TBS buffer containing 0.1% Triton X-100 to eliminate excess primary antibody. Subsequently, primary antibodies were exposed using horseradish peroxidase-conjugated mouse or rat IgG Vectastain ABC kit and DAB/substrate reagents (Vector Laboratories, Burlingame, CA) according to the instructions of the manufacturer. Immunostaining of 10 random cryostat sections (each spaced by at least five unused sections to prevent double counts) of each brain hemisphere (similar regions) was performed on APP/ΔPS1-Tg with each condition and counted. Images were captured using NIS Elements F 3.0 (Nikon), with a Photmetrics CoolSpan cf camera (Tucson, AZ, USA) and a Nikon Eclipse E800 microscope.

### 2.4. Aβ Peptide and NGF Analysis

The brain tissue was weighed and homogenized in 4 vol of PBS buffer (125 mg/mL) containing complete protease inhibitor solution (Sigma-Aldrich, St. Louis, MO, USA). Approximately half volume of the homogenates was combined with 8.2 moL/L guanidine-HCl (pH 7.4) for a final concentration of 5 moL/L and mixed for 4 h at room temperature. The guanidine extracts were diluted with BSAT-DPBS buffer (Dulbecco’s phosphate-buffered saline with 5% bovine serum albumin, 0.03% Tween 20, and 1× protease inhibitor cocktail) at 1:50, mixed, and spun at 16,000× *g* for 20 min at 4°C. The supernatants were analyzed for Aβ peptide levels using human Aβ_1–40_ and Aβ_1–42_ ELISA kits (Biosource International, Camarillo, CA, USA). The remainder of the brain material was centrifuged at 12,000× *g* for 20 min at 4 °C. Brain NGF levels were measured using the Chemikine NGF ELISA kit (Millipore, Billerica, MA, USA) as directed by the manufacturer.

### 2.5. CTFβ and sAβPPα Analyses

C-terminal fragment (CTF) β and sAβPPα were measured by western blot assays as previously described using equal concentrations of protein [22]. CTFβ levels were analyzed in the pellet fraction from the brain extract (antibody 8717, Sigma). Additionally, sAβPPα levels were determined in the supernatant from the brain extracts using the antibody 6E10 (Signet Laboratories, Dedham, MA, USA). CTFβ and sAβPPα levels were determined by densitometry and expressed as percentage of the mean CTFβ and sAβPPα compared to the control group. β-actin was used as a control for the western blots (anti-β-actin from Cell Signaling Technology) and was examined by loading equal amounts of samples (20 µg protein) in the gel lanes.

### 2.6. ELISA Analysis

Quantitative cytokine analysis was determined using enzyme-linked immunosorbent assay (ELISA) to measure the levels of tumor necrosis factor-α (TNF-α), interleukin-1β (IL-1β), or transforming growth factor-β (TGF-β) in brain tissue [23]. Extraction of the cytokines from mouse brains was performed as follows: frozen brains were homogenized in the buffer containing protease inhibitor cocktail (Sigma, St Louis, MO, USA) 1:1000 dilution, and processed using a polytron tissue homogenizer. Tissue suspensions were allocated and snap frozen in liquid nitrogen. Invitrogen ELISA kits were then used, according to manufacturer directions (Carlsbad, CA, USA).

### 2.7. Morris Water Maze Test

An aluminum water tank with a diameter of 100 cm and height of 40 cm was filled with water to a depth of 25 cm at a temperature of approximately 23 °C and a small amount of non-toxic white paint was added to cloud the water. Around the edge of the tank, north (N), south (S), east (E), and west (W) were arbitrarily designated; four alternative start positions were defined by the division of the tank into four quadrants: NE, SE, SW, and NW. A small square white platform (10 × 10 × 2 cm) was submerged to about 1.0 cm below the water surface and placed at the center of the NE quadrant. The platform was generally invisible to the animals. We provided four visible cues to assist the mice in spatial analysis and were placed outside the wall of the pool on the walls. We utilized a camera and SMART video tracking system (San Diego Instruments, San Diego, CA, USA), which was secured approximately 1.6 m above the center of the tank, all trials were recorded. Four months after starting GF treatment, each of the APP/ΔPS1-Tg mice were trained for 4 consecutive days to find the platform from the four different starting points. As previously noted, the starting points were randomly selected every day during the 4 day trials. The mice were gently placed in the pool facing the tank wall. Each mouse was permitted to swim and find the platform within a 60 s period for spatial learning. Once the mouse reached the platform, it was allowed to remain on the platform for 30 s. If the mouse failed to reach the platform, it was placed on the platform for 30 s. Each mouse was allowed to rest for 10 min before the next trial began. The platform was removed on the day after the last training session, and the mice were allowed to search for the platform for an additional 60 s to determine memory retention. For the comparison of spatial learning and memory retention between the mice, the escape latency and distance were determined on the 4th day before removing the platform. Finally, the time that each mouse spent in the NE quadrant (target quadrant) and in the outer annular were assessed on the last day in the absence of the platform.

### 2.8. Statistical Analysis

Results were stated as mean ± standard error of the mean (SEM). Multiple comparison of the number of amyloid plaque, measurements of Aβ peptide (Aβ_1–40_ and Aβ_1–42_) and inflammatory markers were achieved by one-way analysis of variance followed by Tukey’s multiple comparison tests. A two-way analysis of variance (ANOVA) and post hoc of Bonferroni or Tukey test, was used to determine the differences in number of GFAP-positive astrocytes and CD68-positive microglia in the brain. Statistical analysis of the MWM test was determined using a two-way ANOVA. For the analyses, a difference with a *P* value less than 0.05 was deemed statistically significant.

## 3. Results

### 3.1. Impact of GF Diet on Amyloid Pathology in the APP/PS-1 Mice

As previous studies have shown, the APP/PS1 mice used in these experiments started to demonstrate increased levels of A peptides and presence of amyloid plaques in the brain at about 3–4 months of age. This increase of amyloid load or oligomerization was connected with both GFAP-positive astrocytes and CD-68-positive microglia in the cortex area and the hippocampal region [18,24]. Therefore, 3- to 4-month-old APP/PS1 mice display early phases of AD-like pathology and were designated for testing GF diet intervention. Furthermore, these mice show a significant escalation in amyloid load and memory impairment at 6 to 8 months of age [18,24].

Four-month-old APP/PS1 mice were treated with GF diets at 2% and 4% daily (p.o.). Age-matched, vehicle-treated transgenic mice were also used in these experiments. After four months of treatment, all mice were subjected to behavioral analysis (see Section 3.2), euthanized and sections of the brain hemisphere were subjected to immunohistochemical analysis. To understand the impact of GF diets on the levels of Aβ peptides, we measured both Aβ_1–40_ and Aβ_1–42_ peptides by ELISA and the number of amyloid deposits were determined. Immunostaining of 15 random cryostat sections (each spaced by at least five unused sections to prevent double counts) of each brain hemisphere was performed on the mice (Figure 1). As seen in Figure 1, A through C, the number of amyloid plaques was attenuated 4 months after the GF diet treatment (see also Figure 1D–F, higher magnification). Treatment with GF diets at 2 or 4% showed a 54% or 78% decrease in the number of amyloid deposits [vehicle, 254 ± 49.6; GF (2%), 116 ± 21.3; GF (4%), 55.7 ± 26.9] (Figure 1G). Furthermore, the levels of Aβ peptides (Aβ_1–40_ and Aβ_1–42_) in the brain were examined by ELISA (Figure 1H,I). The guanidine-extractable Aβ_1–40_ and Aβ_1–42_ peptides in the brain were decreased by 67% and 87.5% and 52% and 80% in the brain, at the 2 and 4% dosing.

GF diet-treated animals showed a reduction in brain CTFβ (Figure 2A,C). The CTFβ levels were significantly lower (53% and 61%) than in the vehicle animals. These data are also consistent with GF diets mediating the activity of β-secretase since CTFβ is the cleavage product of β-secretase [22]. Mice treated with GF diets also demonstrated an increase in brain sAβPPα (Figure 2B,D). The sAβPPα levels were significantly higher in the GF diet-treated animals (163% and 197%) than the vehicle-treated animals. These data further provide evidence that GF diets are facilitating a diminution of β-secretase activity because sAβPPα is a cleavage product of α-secretase cleavage, which competes with β-secretase activity, and the inhibition of β-secretase activity affords more APP for α-secretase to generate more sAPPα.

### 3.2. Attenuation of Neurological Deficits in APP/PS-1 Mice during GF Diet Treatment

APP/PS1 mice exhibit significant amyloid plaque deposition at 6 to 8 months of age. The mice also show signs of cognitive impairment [18,24,25]. Four months after the GF diets were initiated, the mice were tested for spatial learning and memory retention in the Morris water maze test [18,24]. On the 4th day of training, the vehicle and GF diet-treated animals were examined for the spatial learning ability (escape latency and total distance traveled by mice to reach the platform in the presence of distal visual cues outside the swimming pool). The mice treated with the GF diets displayed a appreciably heightened spatial learning ability in contrast to the vehicle-treated mice (Figure 3A,B). The latency period and distance travelled were 23.1 ± 6.33 s and 101.2 ± 17.2 cm (2%) and 18.6 ± 3.55 s and 69.5 ± 17.8 cm (4%) for the mice treated with GMF diets. For the vehicle mice, the latency period and distance were 52.7 ± 5.34 s and 200.4 ± 20.66 cm, respectively. In the spatial probe (memory retention) trial, when the platform was removed, the GF diet handled mice exhibited significantly lower cognitive impairment levels (better memory retention) when compared to the vehicle-treated mice (Figure 3D,E). The percentage of time spent in the NE quadrant by GF diet-treated mice was 37.5 ± 2.94 (2%) and 41.7 ± 3.91 (4%) seconds compared to 12.6 ± 3.06 s spent by vehicle mice. The percentage of time spent by the groups of mice in the outer annular area were 17.4 ± 4.23 (2%) and 15.3 ± 4.01 (4%) seconds and 43.7 ± 4.52 s (vehicle-treated).

### 3.3. Reduced Inflammation in APP/PS-1 Mice with GF Diets

Next, we examined the potential impact of the Aβ oligomers and amyloid deposits in the APP/PS1 mice on inflammatory mechanisms. We examined the brains using immunohistochemical analysis from vehicle and GF diet-treated mice. The presence of CD68-positive microglial cells was decreased in mice treated with the GF diets (Figure 4B,C) when compared to the vehicle mice (Figure 4A). Quantitative analysis of CD68-positive microglia revealed that the number of cells was reduced in the GF diet-treated mice compared to the control mice, 58% (2%) and 62% (4%) in the brain (Figure 4G) (183.9 ± 31.34 and 166.7 ± 52.47 versus 435.7 ± 66.59). Similarly, the density of GFAP-positive astrocytes was reduced in the APP mouse brains (Figure 4E,F) compared to vehicle-treated APP mouse brains (Figure 4D). The number of GFAP-positive astrocytes was reduced by 59% (2%) and 67% (4%) (389.7 ± 68.33 and 326.9 ± 52.72 versus 956.9 ± 72.81) (Figure 4H). See Appendix A for larger pictures of the brains.

To further understand the role that inflammation plays in the animals treated with GF diets, the APP/PS-1 mouse brain were examined for the expression of inflammatory markers. We measured the levels of the following cytokines: tumor necrosis factor-α (TNF-α), interleukin-1β (IL-1β), and transforming growth factor-β (TGF-β) at the end of the experiment (Figure 5). As seen in the figure, the GF diet-treated mouse brain had significantly reduced TNF-α, IL-1β, and TGF-β levels. The GF diets at 2% and 4% were effective in reducing the cytokine levels by 59% and 78% (TNF-α), 78% and 89% (IL-1β), and 82% and 91% (TGF-β).

### 3.4. Reduced Cathepsin B Expression in GF Diet-Treated APP/PS-1 Mice

Previous studies have implicated cathepsin B as both a β-secretase and part of the inflammasome complex [19,26]. We have shown that in APP mice, cathepsin B is elevated and influences Aβ peptide generation and inflammation [18,19]. Our studies have shown that elevated cathepsin B protein and activity can lead to expression of inflammatory mediators such as IL-1β [18,19]. To determine the role of GF diets on the increase in inflammation (Figure 6), we established the impact on cathepsin B protein levels. GF diets decreased cathepsin B levels [61% (2%) and 85% (4%)] in the brain, https://www.ncbi.nlm.nih.gov/pmc/articles/PMC3934599/figure/f1/ suggesting that reduction in inflammation occurring with treatments was partially related to the inhibition of cathepsin B activity. In addition, we have shown (Figure 1) that Aβ peptide levels are decreased in GF diet-treated animals.

### 3.5. Increased Expression of NGF in GF Diet-Treated Mice

Finally, we determined the impact of GF diets on the levels of NGF in brain tissue extracts from the APP/PS-1 mice (Figure 7). We detected that the APP/PS-1 GF diet-treated mice had significantly elevated concentrations of NGF in the brain than APP/PS-1 vehicle-treated mice [42.3 ± 6.52 (2%) and 71.4 ± 9.72 (4%) vs. 5.41 ± 2.11 pg/mg). We found that the GF diets rendered an eight- to twelve-fold increase of NGF content in the brain, and that the increase was highly significant (*p* < 0.001).

## 4. Discussion

In the present work, we show that diets enriched in anti-inflammatory agents and antioxidants can limit the extent of Alzheimer’s disease (AD). We show that continuous treatment at the onset of symptoms/pathology significantly ameliorates the clinical course of the disease. The effect parallels a decrease in both inflammation and Aβ peptides in the affected areas of the brain. Moreover, decreases in CTFβ and increases in sAPPα, which are consistent with changes in processing enzymes, were detected in the brains of the treated mice [18,19]. In addition, our in vivo studies also demonstrate that cathepsin B levels are attenuated by the diets, which may influence the changes in inflammation and Aβ peptide levels detected.

Alzheimer’s disease (AD) is an age-related neurodegenerative disease that results in progressive memory deficits [27]. The disease is characterized by increases in inflammation, reactive oxygen species, and other manifestations and culminate in a number of neurological conditions leading to death [28]. AD accounts for about 70% of all causes of dementia and is among the leading causes of death in the elderly [29,30]. AD is complex, and it is unlikely that any one drug or other intervention will successfully treat it. Current approaches focus on helping people maintain mental function, manage behavioral symptoms, and slow down the symptoms of disease [31]. The currently approved therapeutics are Razadyne^®^ (galantamine), Exelon^®^ (rivastigmine), and Aricept^®^ (donepezil), which help to improve memory, but do not have an impact on the disease process [32].

AD is distinguished by the histological presence of A peptides, which accumulate into intracellular fibrils and extracellular plaques, and by the existence of additional intracellular neurofibrillary tangles (NFTs) comprised of hyperphosphorylated tau protein [33]. Even though we have an extensive database of information, how these two entities are involved in the disease process is still a work in progress [34]. The impact of A and tau on cellular mechanisms related to neuronal cell dysfunction, shrinkage, and loss are still being examined, and a multitude of avenues are being pursued [35]. The amyloid hypothesis implicates the APP protein and processing by β- and γ-secretases that alter neuronal function, disrupts synaptic activity, and can trigger apoptosis [36]. There is debate on the eliciting of NFTs formation, which is either a result of Aβ oligomers stimulating tau hyperphosphorylation or direct effects of the disease process [37]. An inequity in neuronal production or glial/glymphatic clearance of the Aβ peptides can result in the formation of oligomers or aggregates that can have neurotoxic effects [38]. The relationship between aging/inflammation and generation of Aβ/tau in the brain and cyclic effects of both is gaining momentum, and the impact of the inflammasome is of significant interest [39].

Inclusion of diets enriched in anti-inflammatory and antioxidant agents, and demonstrated a significant reduction in inflammation. Previous experiments in mouse models of AD inflammation comes across as a key pathway in the mechanisms of the disease [19,40]. We and others have shown the specific cytokines and chemokines, as well as inflammatory modulators, are important in the pathogenesis of AD [19,41,42]. In addition, increases in CD-68 and GFAP which are markers of microglia and astrocytes are seen in neurological diseases and may contribute to the inflammation and disease [43]. However, therapeutics to limit the impact of inflammation and oxidative stress have not been successful in treating neurodegenerative diseases like AD [44]. Diets enriched in natural sources of anti-inflammatory agents and antioxidants may provide an alternative to targeted approaches [15]. Our data shows that the diets can function to limit activation of astrocyte and microglia as well as alter cathepsin B (catB) activity [16]. Previous studies have implicated catB in activation of the inflammasome and release of IL-1β as well as functions as a β-secretase in the generation of Aβ from APP [19]. These data suggest that the diets can reduce Aβ peptide levels and IL-1β to slow the disease process down.

## 5. Conclusions

The data presented here show that GF diets can alter the pathogenesis of AD when administered early in the disease process. Alteration in diet, which will reduce inflammation and reactive oxygen species, may affect the development and progression of AD, and acute and chronic treatments may provide a therapeutic impact on AD and may attenuate the disease process. These studies demonstrate that diets enriched in phytonutrients may have beneficial effects in AD and other neurodegenerative disorders.

## Figures and Tables

**Figure 1 nutrients-13-00117-f001:**
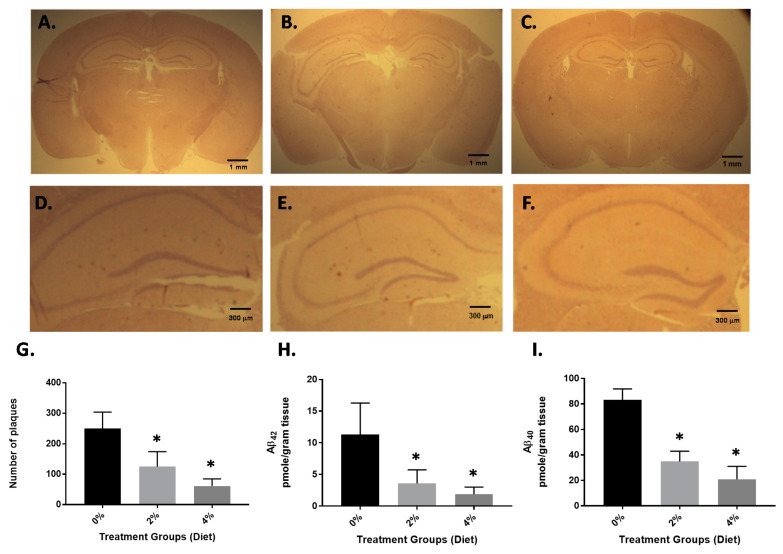
Amyloid load in APP/PS1 mice after 4 months of GF diet treatment. Brain hemisphere sections obtained from control mice (**A**) GF diet treated with 2% (**B**) or GF diet treated with 4% (**C**) were immunostained with mouse anti-human Aβ peptide (clone 10D5) antibody to detect amyloid plaques. Higher magnification (**D**–**F**). The number of amyloid plaques in the brain sections (15 sections per mouse) from each set of control mice or mice treated with GF diets were counted and averaged (**G**). The brain hemispheres were examined for guanidine-extractable Aβ_1–40_ (**H**), and Aβ_1–42_ peptides (**I**). Vehicle: 8-month-old APP/PS1-Tg mice that were not treated with GF diets were used as controls (* *p* < 0.001, *n* = 20 per group).

**Figure 2 nutrients-13-00117-f002:**
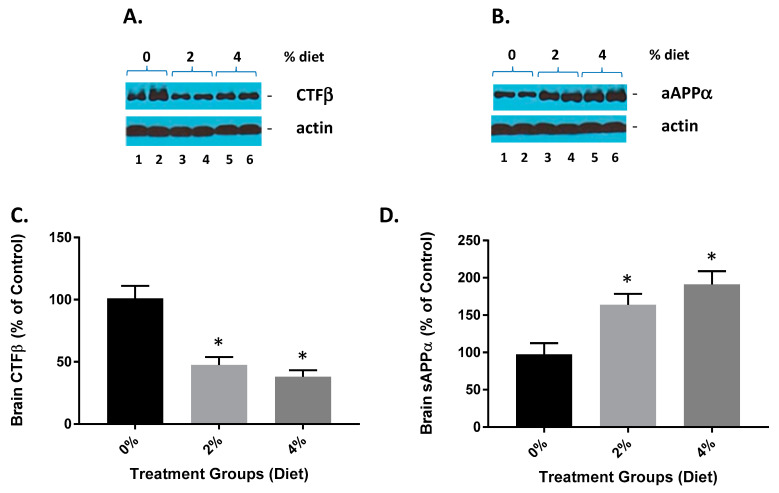
GF diets lower and increase brain CTFβ and sAβPPα. (**A**) GF diets caused a reduction in brain CTFβ. (**B**) GF diet mediated increase in brain sAβPPα levels. (**C**,**D**). Quantification of blots from (**A**,**B**), respectively. These data are consistent with GF diets mediating the inhibition of β-secretase activity because sAβPPα is derived from AβPPα-secretase cleavage, which competes with βsecretase cleavage of AβPP, and thus the inhibition of β-secretase activity provides more AβPP for β-secretase to produce more sAβPPα (* *p* < 0.001; *n* = 20).

**Figure 3 nutrients-13-00117-f003:**
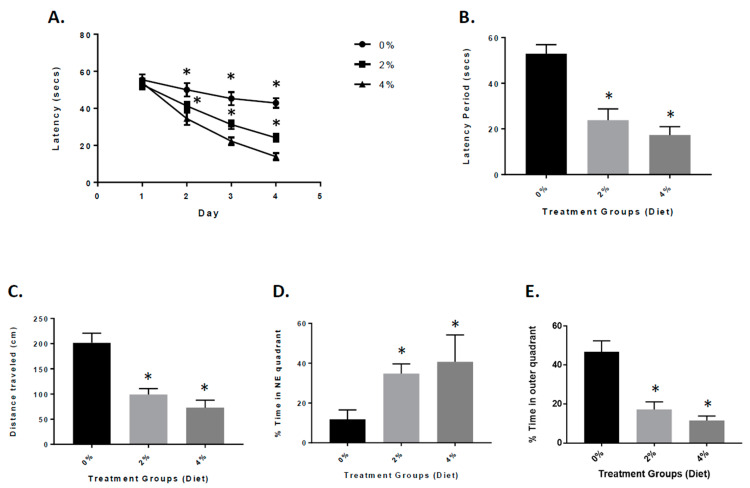
Behavioral analysis of mice treated with GF diets in the Morris water maze. (**A**) APP/PS-1 vehicle, 2% and 4% GF diets were examined for memory acquisition. APP/PS-1 mice were trained in the Morris water maze test on each of 4 consecutive days to learn the location of a submerged, invisible platform in a pool of water. The time that it took the mice to swim to the platform was recorded each day, measured as the latency period (seconds), with shorter latency times indicating better memory acquisition. Latency (secs) is shown as x ± s.e.m. (* Statistically significant, *p* < 0.05, *n* = 20 per group). Memory deficits of APP/PS-1 mice were assessed 2 days after completion of the training in the Morris water maze test by measuring the latency period (**B**) and distance traveled (**C**) for animals to swim to the submerged, invisible platform. The shorter latency periods and shorter distances traveled indicate improved memory. Values are expressed as x ± s.e.m., and *n* = 20 per group. *Statistically significant (*p* < 0.001). The percent time each animal swam in the quadrant from which the platform had been removed (Northeast (NE) quadrant (target quadrant), (**D**) and the percent time an animal swam in the annulus of the pool were recorded (**E**). Values are expressed as the mean ± s.e.m., and *n* = 20 per group. * Statistically significant with *p* < 0.001.

**Figure 4 nutrients-13-00117-f004:**
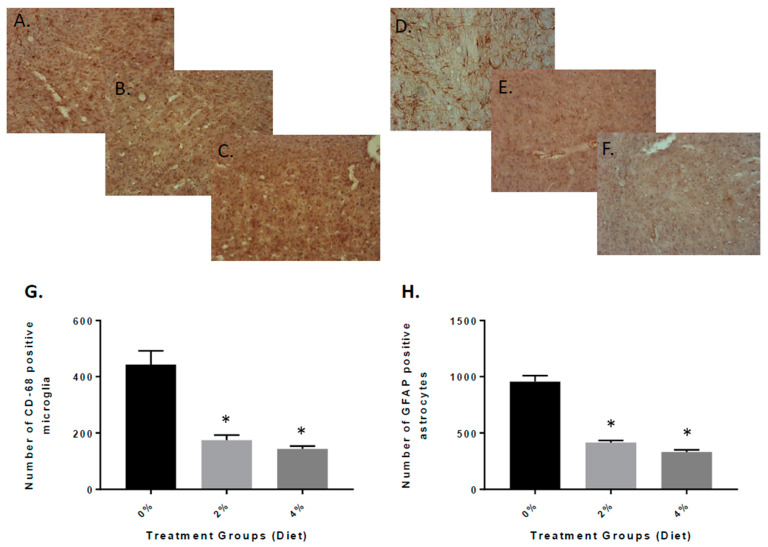
Quantitative analysis of activated microglia and astrocytes in the brain of APP/PS1 mice after 4 months of GF diet treatment. Brain sections from APP/PS1 control mice (**A**) or mice treated with GF diet at 2% (**B**) or 4% (**C**) were stained with anti-mouse CD68 antibody to detect the activated microglial cells. Ten sections of each mouse were counted and averaged for the number of CD68-activated microglia in the brain (* *p* < 0.001, *n* = 20). Brain sections from APP/PS1 control mice (**D**) or mice treated with GF diet 2% (**E**) or 4% (**F**) were immunostained with anti-mouse GFAP antibody to detect the activated astrocytes. Ten sections of each mouse were counted and averaged for the number of GFAP-activated astrocytes in the brain (* *p* < 0.001; *n* = 20). (**G**,**H**): quantification of data from the section for CD-68 and GFAP, respectively.

**Figure 5 nutrients-13-00117-f005:**
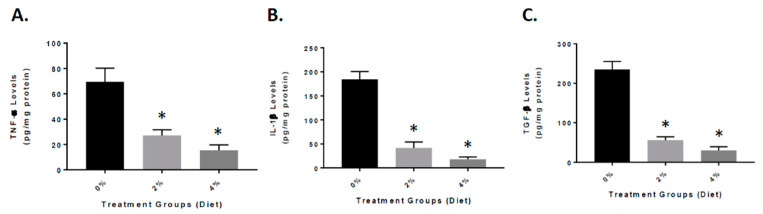
Reduced inflammatory markers in the brain of APP/PS-1 mice. Mice were treated with vehicle (Vehicle), or GF diets for 4 months at 2% or 4%. Quantitative analysis of TNF-α (**A**), IL-1β (**B**), and TGF-β (**C**) in the APP/PS-1 brains was determined by enzyme-linked immunosorbent assay (ELISA). *n* = 20 per group. * *p*  <  0.01 (compared with vehicle).

**Figure 6 nutrients-13-00117-f006:**
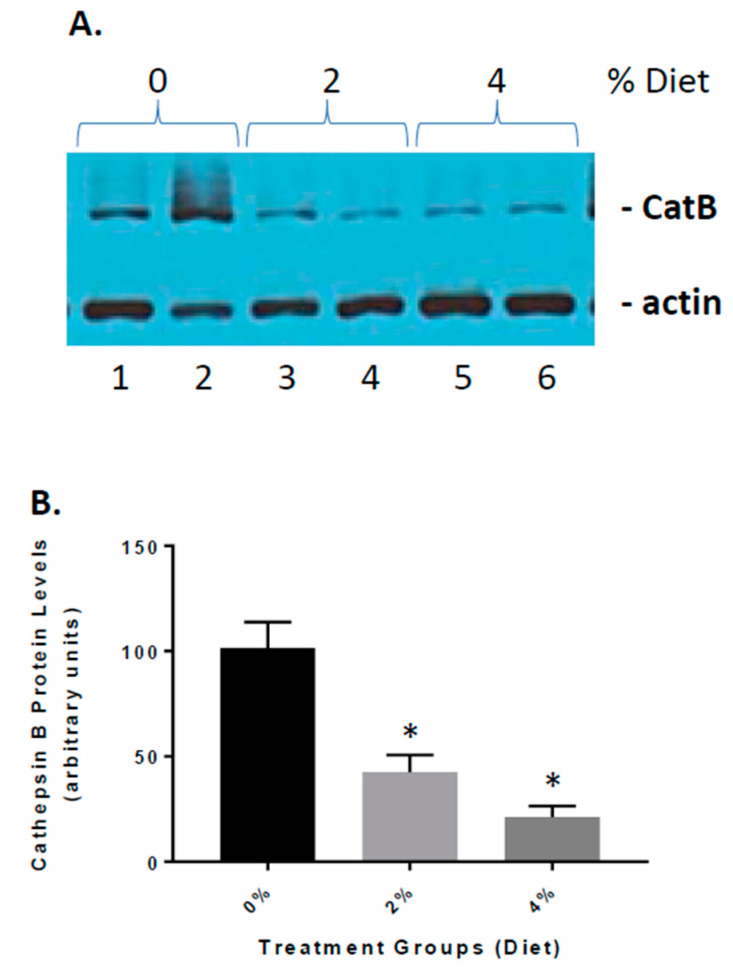
The effect of GF diets on cathepsin B activity. (**A**) Brain cathepsin B protein levels were determined at the end of the experiment. Western blot analysis of the cathepsin B levels in the brains of vehicle, 2% and 4% GF diet-treated mice. (**B**) Quantitative analysis of cathepsin B protein levels of the mice in A. The results are expressed as mean +/− SEM (*n* = 20; * *p* < 0.001 compared to the vehicle group).

**Figure 7 nutrients-13-00117-f007:**
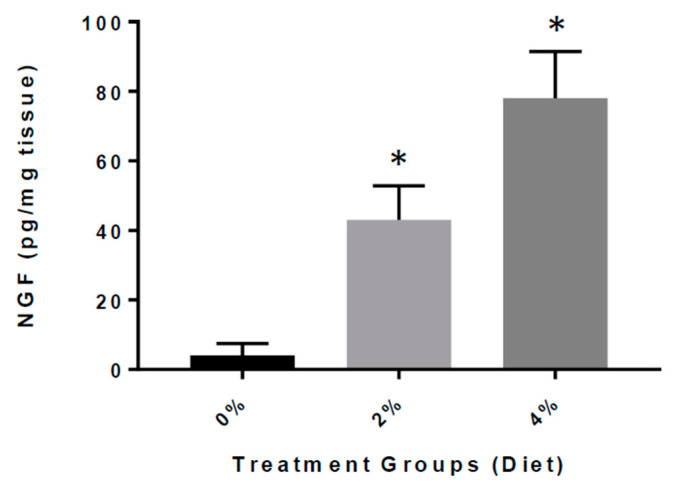
Concentration of NGF in the brain of APP/PS-1 mice. Bar graphs show the concentration of NGF in picograms (pg) per milligram (mg) of brain tissue from APP/PS-1 mice treated with GF diets (2 or 4%) versus APP/PS-1 mice untreated. The difference was statistically significant (* *p* < 0.001; *n* = 20 per group).

## Data Availability

The data presented in this study are available on request from the corresponding author. The data are not publicly available due to privacy issues.

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
