# Peer review of "Effects of GrandFusion Diet on Cognitive Impairment in Transgenic Mouse Model of Alzheimer’s Disease"

_nutrients, 2020, doi:10.3390/nu13010117_

Round 1
Reviewer 1 Report
Review of Nutrients “effects of diet on cognitive impairment in transgenic mouse model of AD” yu et al.
Summary:
The authors are interested in the effect of the fruit and vegetable based dietary supplement, Grand Fusion (GF) on prevention or treatment of AD. They treat APP/ΔPS1 mice from 4-8 months with daily gavage of 2% or 4% GF or vehicle. At the end of treatment, learning and memory are tested by morris water maze, and mice treated with diet performed better than mice treated with vehicle. In addition, mice treated with GF diet had reduced amyloid plaques, and Abeta as measured by ELISA. They also report decreased b-CTF and increased a-CTF, suggesting a change in APP processing. They hypothesize that the diet mediates its effect through antioxidant/anti-inflammatory mechanisms, so they look at cytokines, and find that inflammatory cytokines are reduced, as are activated astrocytes and microglia. Lastly, NGF is elevated by the diet, while cathepsin B is reduced.
Overall impression:
The research is interesting, and appears to be well done, though some methods need better explanation to verify that.
Questions/problems:
- The symbols for α, β, and other Greek symbols are often not showing up correctly, making some sentences hard to interpret. This should be corrected
- A few language issues, wrong words used:
Pg 2, line 50: where for were
Pg 3 line 118: weighted for weighed
Pg 4, line 178: elaborate for demonstrate??
Pg 5 line 192: tempered for decreased?? And interconnected for ??
- Page 4 line 154 states “four months after viral injection …” but there is no mention anywhere else of viral injection. This was probably copied from another publication and not edited out?
- The resolution of the images in Figure 1A-C and Figure 4A-F is poor. When the PDF is magnified to see them, they are very blurry, so the changes in plaque number and microglia and astrocytes cannot be seen.
- In addition, there is no information in the methods regarding how the images were captured, then how the counting of plaques and microglia and astrocytes was done. Since the images are not good, it is hard to see what might have been counted, and the IHC data is not convicing. Please add methods regarding microscopy and counting methods. Were they hand counted, or was it automated in a program? What regions were counted, or was it the whole brain or hemibrain in the coronal section? Were these sections in approximately the same bregma position(s) for each animal? Due to the extensive processes in microglia and astrocytes, it can be hard to accurately count individual cells; % area covered is another common measure.
- Also, in all figures, it is not clear how many mice were used for each group to generate the data. The only mention of animal numbers is that a total of 120 animals were used (pg 2, line 75) and then that vehicle and treatment groups each had 20 mice (pg 2, line 87). So does this mean that all twenty mice were used for the imaging experiments, and western blots and ELISAs? Please include in each figure legend the number of mice per group, and male/female distribution.
- The shift from beta to alpha cleavage of APP is interesting (Fig 2). Did you look to see if levels of BACE1 or alpha-secretase were changed, or other ways this could have happened?
- In Figure 3 legend, it is conventional to refer to quandrant with platform as “target quadrant” rather than the directional location. That way the reader understands without having to have the layout of the specific experiment explained.
- Page 9, line 290- There is no strong evidence for cathepsin B as the beta-secretase, if that is what is meant there. It may play other roles in AD, but is not the beta-secretase
Author Response
- Symbols were corrected.
- Corrected words
- Corrected verbiage for viral comments
- Replaced images
- Information on plaque counts was added to the methods section
- The number of mice per group is indicated in the figure legends, 20 mice per group were used for all analyses. Male/female ratio is indicated in the methods section.
- We did not measure the BACE1 levels or activity in this study
- Target quadrant was added
- We and others have shown that cathepsin B is a better BACE than BACE1 actually is and have indicated in the publications
Reviewer 2 Report
An interesting scientific topic, regarding a devastating disease as Alz's disease.
A lot of data showing the great impact of the diet on the expression of the disease, both clinically and paraclinically.
Please correct: line 365: .... and may attenuate
line 366: ... may have beneficial effects.
Author Response
- Corrected
- Corrected
Reviewer 3 Report
1) The GF diet is capable of significantly attenuating amyloid plaque load, AB peptide levels, cathepsin B levels (inflammation), and increased brain NGF compared to its control transgenic mice (ABPP-PS1). These data are considerable in suggesting that this type of diet enriched in phytonutrients, rich in antioxidants and anti-inflammatories, is beneficial for its ability to slow down the processes of Alzheimer's disease in these mice, and possibly extrapolated in man.
2) Please, whenever you add an abbreviation for the first time, indicate its description first (either in abstract or in text). For example: NGF, MCI, CTF, …
3) Throughout the text of the PDF I see symbols that substitute @ for the Greek letter beta. Please check if they are formatting errors. If you have it correct, ignore it. They can be foreign problems with certain pdf readers.
4) Missing Table 1 (Datos Suppl)
5) In section 2.4. it only measures peptide AB and NFG, and does not indicate it in the title. You have to delete APP and CTF, in section 2.4. CTF and sABPP analysis is not described (you do this in section 2.5.).
6) Line 154: What viral injection? You do not specify that you have built a virus.
7) Figure 2 (A and C). Indeed, GF diets (2% and 4%) reduce brain CTF. If you look closely at lanes 5 and 6 with the 4% diet, the blots may not be representative of what appears in the bar chart. It seems that in the 4% diet there is a greater expression than in the 2% diet, close to 50% of the control.
8) Line 262: Figure 4D would be figure 4G. The same goes for Figure 4F, it would be 4E.Figure 4G would be 4F.Figure 4E would be 4D.The foot of figure 4 is the same.Correct the identifications: (E) would be (D).((F) would be (E), (G) would be (F), and (D) would be (G).
9) Line 281: A repeated % symbol appears. Remove repetition.
10) Line 303: Standard deviation (SD) or error of the mean (SEM)?
11) Line 441: Reference number 25 does not appear in the text.
Author Response
- Corrected abbreviations
- Corrected Greek lettering
- Table 1 added
- Corrected
- Removed viral comments
- Added new figure
- We corrected G for D, the others are correct and are referring to the images and not the graph.
- Removed extra %
- Changed SD to SEM
- Reference 25 was added to the text